# Performance of the Access Bio/CareStart rapid diagnostic test for the detection of glucose-6-phosphate dehydrogenase deficiency: A systematic review and meta-analysis

Benedikt Ley[1]*, Ari Winasti Satyagraha[2], Hisni Rahmat[2], Michael E. von Fricken[3], Nicholas M. Douglas[1], Daniel A. Pfeffer[1], Fe Espino[4], Lorenz von Seidlein[5,6], Gisela Henriques[7], Nwe Nwe Oo[8], Didier Menard[9], Sunil Parikh[10], Germana Bancone[6,11], Amalia Karahalios[12], Ric N. Price[1,5,6]

1 Global and Tropical Health Division, Menzies School of Health Research and Charles Darwin University, Darwin, Australia, 2 Eijkman Institute for Molecular Biology, Jakarta, Indonesia, 3 Department of Global and Community Health, George Mason University, Fairfax, Virginia, United States of America, 4 Research Institute for Tropical Medicine, Department of Health, Muntinlupa City, Philippines, 5 Mahidol-Oxford Tropical Medicine Research Unit (MORU), Faculty of Tropical Medicine, Mahidol University, Bangkok, Thailand, 6 Centre for Tropical Medicine and Global Health, Nuffield Department of Clinical Medicine, University of Oxford, Oxford, United Kingdom, 7 Faculty of Infectious and Tropical Diseases, London School of Hygiene & Tropical Medicine, London, United Kingdom, 8 Department of Medical Research (Lower Myanmar), Yangon, Republic of the Union of Myanmar, 9 Malaria Genetics and Resistance Unit, Institut Pasteur, Paris, France, 10 Department of Epidemiology of Microbial Diseases, Yale School of Public Health, New Haven, Connecticut, United States of America, 11 Shoklo Malaria Research Unit, Mahidol-Oxford Tropical Medicine Research Unit, Faculty of Tropical Medicine, Mahidol University, Mae Sot, Thailand, 12 Centre for Epidemiology and Biostatistics, Melbourne School of Population and Global Health, The University of Melbourne, Melbourne, Australia

* benedikt.ley@menzies.edu.au

## Abstract

### Background

To reduce the risk of drug-induced haemolysis, all patients should be tested for glucose-6-phosphate dehydrogenase (G6PD) deficiency (G6PDd) prior to prescribing primaquine (PQ)-based radical cure for the treatment of vivax malaria. This systematic review and individual patient meta-analysis assessed the utility of a qualitative lateral flow assay from Access Bio/CareStart (Somerset, NJ) (CareStart Screening test for G6PD deficiency) for the diagnosis of G6PDd compared to the gold standard spectrophotometry (International Prospective Register of Systematic Reviews [PROSPERO]: CRD42019110994).

### Methods and findings

Articles published on PubMed between 1 January 2011 and 27 September 2019 were screened. Articles reporting performance of the standard CSG from venous or capillary blood samples collected prospectively and considering spectrophotometry as gold standard (using kits from Trinity Biotech PLC, Wicklow, Ireland) were included. Authors of articles

**Data Availability Statement:** All data included in the submission can be obtained from the source

articles directly. Contact details for data requests are provided in S3 Table.

**Funding:** No specific funding was received for this study. However, BL is funded through a fellowship from the Menzies School of Health Research, Darwin, Australia. BL and RNP are funded by the Australian Department of Foreign Affairs and Trade (74431) and the Bill & Melinda Gates Foundation (OPP1054404 and OPP1164105). RNP is also funded by the Wellcome Trust (Senior Fellowship in Clinical Science, 200909) and the Australian Centre for Research Excellence on Malaria Elimination (APP 1134989). No funding bodies had any role in study design, data collection and analysis, decision to publish, or preparation of the manuscript.

**Competing interests:** I have read the journal's policy and the authors of this manuscript have the following competing interests: LvS receives a stipend as a Specialty Consulting Editor for *PLOS Medicine* and serves on the journal's editorial board. All other authors declare no competing interests.

**Abbreviations:** AMM, adjusted male median; AUC, area under the curve; CSG, CareStart Screening test for G6PD deficiency; FN, false negative; FP, false positive; FST, fluorescent spot test; G6PDd, glucose-6-phosphate dehydrogenase deficiency; Hb, haemoglobin; IPD, individual participant data; IQR, interquartile range; LR+, positive likelihood ratio; LR−, negative likelihood ratio; NPV, negative predictive value; PPV, positive predictive value; PQ, primaquine; PRISMA, Preferred Reporting Items for Systematic reviews and Meta-Analyses; PROSPERO, International Prospective Register of Systematic Reviews; QUADAS, Quality Assessment of Diagnostic Accuracy Studies; RBC, red blood cell; SROC, summary receiver operator characteristic; TN, true negative; TP, true positive; TQ, tafenoquine.

fulfilling the inclusion criteria were contacted to contribute anonymized individual data. Minimal data requested were sex of the participant, CSG result, spectrophotometry result in U/gHb, and haemoglobin (Hb) reading. The adjusted male median (AMM) was calculated per site and defined as 100% G6PD activity. G6PDd was defined as an enzyme activity of less than 30%. Pooled estimates for sensitivity and specificity, unconditional negative predictive value (NPV), positive likelihood ratio (LR+), and negative likelihood ratio (LR−) were calculated comparing CSG results to spectrophotometry using a random-effects bivariate model.

Of 11 eligible published articles, individual data were available from 8 studies, 6 from Southeast Asia, 1 from Africa, and 1 from the Americas. A total of 5,815 individual participant data (IPD) were available, of which 5,777 results (99.3%) were considered for analysis, including data from 3,095 (53.6%) females. Overall, the CSG had a pooled sensitivity of 0.96 (95% CI 0.90–0.99) and a specificity of 0.95 (95% CI 0.92–0.96). When the prevalence of G6PDd was varied from 5% to 30%, the unconditional NPV was 0.99 (95% CI 0.94–1.00), with an LR+ and an LR− of 18.23 (95% CI 13.04–25.48) and 0.05 (95% CI 0.02–0.12), respectively.

Performance was significantly better in males compared to females ($p = 0.027$) but did not differ significantly between samples collected from capillary or venous blood ($p = 0.547$). Limitations of the study include the lack of wide geographical representation of the included data and that the CSG results were generated under research conditions, and therefore may not reflect performance in routine settings.

## Conclusions

The CSG performed well at the 30% threshold. Its high NPV suggests that the test is suitable to guide PQ treatment, and the high LR+ and low LR− render the test suitable to confirm and exclude G6PDd. Further operational studies are needed to confirm the utility of the test in remote endemic settings.

## Author summary

### Why was this study done?

- Glucose-6-phosphate dehydrogenase (G6PD) deficiency (G6PDd) is the key determinant of severe haemolysis following primaquine (PQ)-based radical cure of vivax malaria.

- A widely available reliable point-of-care diagnostic for G6PDd will improve patient safety of PQ treatment.

- A rapid diagnostic G6PD test from Access Bio (Somerset, NJ) has operational characteristics that render the test suitable for use at the bedside.

## What did the researchers do and find?

- We reviewed the literature systematically and identified studies that had evaluated the G6PD test and compared results with those generated by the gold standard spectrophotometry.

- Individual participant data (IPD), available from 5,777 participants, demonstrated that the test had a 96% sensitivity for detecting G6PD-deficient individuals with a specificity of 95%.

## What do these findings mean?

- Under research conditions, the G6PD test reliably confirms and excludes G6PDd in patients with G6PD activity of less than 30% (the most widely applied cut-off activity to guide PQ-based radical cure).

- These findings will have to be confirmed in routine clinical settings.

## Introduction

Radical cure of *Plasmodium vivax* and *P. ovale* malaria requires killing of both the blood and liver stages of the parasite to prevent relapsing malaria and reduce ongoing transmission [1]. Primaquine (PQ) has been used for over 65 years and is currently the only widely available hypnozoitocidal drug for *P. vivax* and *P. ovale*. PQ has to be administered in combination with a blood schizontocidal agent over 7 to 14 days to clear hypnozoites [2–6]. While PQ is tolerated in most patients, it can cause haemolysis in patients with glucose-6-phosphate dehydrogenase deficiency (G6PDd), the severity of which is dependent on the underlying genetic variant, the dose of PQ administered, and the age of the patient's red blood cell (RBC) population [7,8].

To date, 215 genotypes conferring different degrees of G6PDd have been described, and these are most prevalent in areas of past and present malaria endemicity [9–11]. The G6PD gene is located on the X chromosome (Xq28), therefore males are either hemizygous G6PD deficient or G6PD normal, whereas females can be homozygous G6PD deficient, G6PD normal, or heterozygous for the gene. In heterozygous females, one copy of the G6PD gene is randomly inactivated through a process called lyonization; accordingly, heterozygous females harbour 2 distinct groups of RBCs, a G6PD normal and a G6PD-deficient one [12]. Depending on the ratio of G6PD-normal to G6PD-deficient RBCs, heterozygous females may be at a risk of severe drug-induced haemolysis [13,14].

To reduce the risk of drug-induced haemolysis, WHO recommends that patients be tested routinely for G6PDd prior to administration of PQ-based radical cure [4]. The gold standard method for measuring G6PD activity is quantitative spectrophotometry [15,16], but this method is expensive and requires laboratory facilities that are often unavailable in malaria-endemic communities, especially in remote areas. The fluorescent spot test (FST) is a qualitative alternative; however, it also requires laboratory infrastructure and extensive training for reliable interpretation [17,18]. In 2011, Access Bio (Somerset, NJ) introduced a qualitative, lateral-flow point-of-care assay (CareStart screening test for G6PDd; CSG) [19]. The aim of this article was to undertake a meta-analysis of published studies to determine the performance of

the assay in a variety of populations at risk of drug-induced haemolysis (International Prospective Register of Systematic Reviews [PROSPERO]: CRD42019110994).

## Methods

### Search strategy and eligibility criteria

A PubMed search was undertaken for relevant articles published in English between 1 January 2011, when the test was first introduced [19], and 27 of September 2019. The search terms applied were "G6PD AND (rapid diagnostic test OR carestart)". Identified articles were first screened for eligibility by title, abstract, and then by the full text by 3 study authors (BL, AWS, and HR) independently. Reference sections of identified articles were screened for additional relevant articles. Eligible articles reported performance indicators of the CSG from samples collected prospectively. Articles describing prototypes of the CSG were excluded. Only studies comparing the CSG results to the gold standard spectrophotometry, using kits from Trinity Biotech PLC (Wicklow, Ireland), were included. Studies were included irrespective of whether blood was collected from capillary or venous sampling.

Corresponding authors of identified articles were contacted and asked to provide anonymized individual participant data (IPD). All corresponding authors were contacted a minimum of 3 times before the study was excluded. Minimal data requested included the sex of the participant, CSG result, spectrophotometry result in U/gHb, and corresponding haemoglobin measurement in U/dL. Data were entered into a customized Excel database (Microsoft Corporation, Redmond, WA) and analysed using Stata software version 14 (release 14; StataCorp, College Station, TX). Analysis was done primarily using the Midas package.

### Data preparation

Invalid CSG results were excluded from the analysis. Spectrophotometry results that were missing or extreme (>25 U/gHb) were excluded from analyses because these readings suggested a procedural or data error. Some studies reported an intermediate CSG result; in clinical use, these are more likely to be considered G6PDd results and were defined accordingly. One article reported the results of 2 separate evaluation studies from Laos and Cambodia [20]; because the applied cut-off activities and reported performance were distinct for each country, the results are reported separately.

The adjusted male median (AMM) was calculated from spectrophotometry results separately for each study site and defined as 100% G6PD activity [16]. Because some studies applied different definitions of 100% G6PD activity (for example, by considering genotype [21]), the definitions within this study and the original source articles may sometimes differ. Studies reported spectrophotometry results either from venous and/or capillary blood, and the source of blood could have affected spectrophotometry measurements. One study measured G6PD activity in paired capillary and venous samples by spectrophotometry [21], and the results were compared for significant differences using the Wilcoxon signed-rank test.

Spectrophotometry provides a quantitative result; following the current informal cut-off to guide PQ-based radical cure [15], and the intended cut-off of the CSG [17,20], any sample with less than 30% of the AMM was defined as G6PDd. Study-specific performance was calculated following standard formulae [16,22,23], by comparing the CSG against the reference method spectrophotometry. A positive result was defined as a G6PDd outcome and a negative result as a G6PD normal outcome. Results from the CSG were then classified as true positive (TP), true negative (TN), false positive (FP), and false negative (FN) with reference to the results of spectrophotometry.

## Data analysis

To calculate the pooled estimates for sensitivity and specificity, a 2-level model with independent binomial distributions was fitted for the TPs and TNs conditional on the sensitivity and specificity in each study, and a bivariate normal model for the logit transformations of the sensitivity and specificity between the studies was created [24]. A summary receiver operator characteristic (SROC) curve was constructed, and the area under the curve (AUC) was calculated to determine overall test performance.

Unconditional predictive values were calculated for G6PDd prevalence of 5% to 30% reflecting G6PDd prevalence within most malaria-affected populations [25]. Likelihood ratios are a convenient method to determine the usability of a diagnostic test. In the case of the CSG, the positive likelihood ratio (LR+) describes how many times more likely a G6PDd test result is to occur in a G6PD-deficient individual compared to in a G6PD-normal individual. The negative likelihood ratio (LR−) is defined as the inverse of this, or how much less likely a G6PD-deficient result will occur in a G6PD-normal person compared to a G6PD-deficient individual [26]. In general, tests with an LR+ above 10 are considered suitable for the diagnosis of a condition, and an LR− of less than 0.1 is considered suitable to exclude a condition [27]. The LR+ and LR− were calculated, and the practical utility of the CSG was evaluated by constructing likelihood ratio diagrams. The quality of the included publications was assessed using the QUADAS-2 tool [28].

## Model validation

$I^2$ was calculated as a measure of heterogeneity for sensitivity and specificity. Publication bias was assessed by a funnel plot, and a linear regression model was fitted to the log odds ratio of the inverse root of effective sample sizes as a test for funnel plot asymmetry.

## Sensitivity analyses

We tested whether the sensitivity and the specificity of the tests varied by type of blood collected (capillary or venous) and sex by fitting separate multilevel models. In the first, we included a covariate for blood type and allowed both sensitivity and specificity to vary by blood type; we then repeated the analysis by instead including a covariate for sex. Each of these models was compared to a model without covariates using a likelihood ratio test. Additional sensitivity analyses were undertaken in which the pooled performance was recalculated excluding studies that were at high risk of bias due to participant selection or laboratory methods. The pooled performance was recalculated applying a pooled AMM across all included studies rather than the study-specific AMM. In response to a reviewer's request, the analysis was repeated including all data as well as the aggregated data extracted from eligible articles for which individual patients' data were not available. The definition of TP, TN, FP, and FN for articles in which no IPD were available was based on definitions applied in the respective studies. Whenever a discrepancy between reported performance and numbers of TPs and FPs and TNs and FNs was found, the latter was considered.

## Results

### Identified studies and participants

A total of 42 articles were identified in the literature review, of which 11 met the inclusion and exclusion criteria. Individual data were available from 8 studies (Fig 1) enrolling a total of 5,815 participants with paired CSG and spectrophotometry measurements (S1 Table).

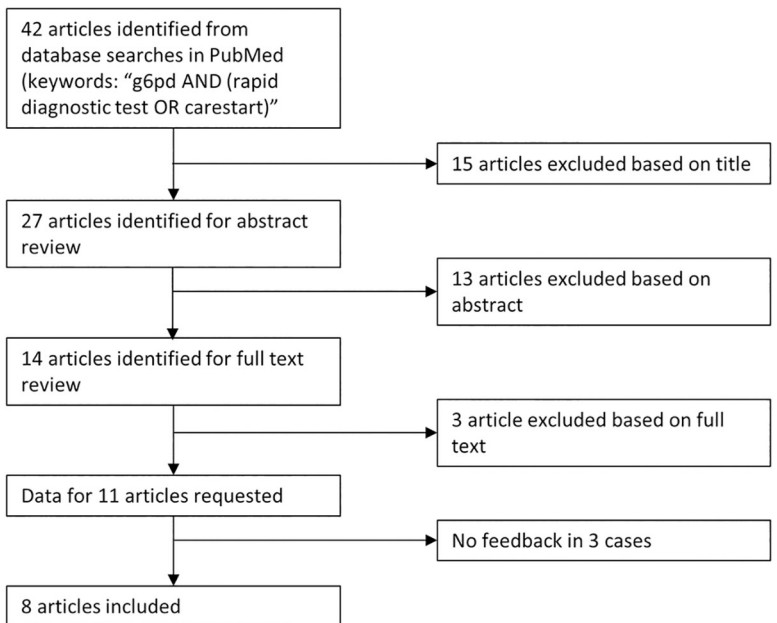

**Fig 1. Flow chart on article selection.**

All of the studies included were undertaken between 2014 and 2018. Six studies were conducted in Southeast Asia [20,21,29–32], one in Africa [33], and one in the Americas [34]. In total, 3 studies (4 countries, 2,845 participants) assessed G6PD status from capillary blood [20,31,33] and 3 from venous blood (3 countries, 2,066 participants) [30,32,34]. In 1 study, CSG and spectrophotometry were performed on both venous and capillary samples [21], and in 1 study CSG was performed on both venous and capillary samples; however, spectrophotometry was only performed on capillary blood [29] (Table 1). Results from 14 (0.2%)

**Table 1. Origin, source of blood, and results included.**

| Article | Blood | Country | Original sample size | G6PD > 25 U/gHb or missing (%) | Invalid CSG result (%) | Total included (%) |
|---|---|---|---|---|---|---|
| Bancone, 2015* [21] | Capillary | Thailand | 150 | 0 (0.0) | 12 (8.0) | 138 (92.0) |
| Bancone, 2015* [21] | Venous | Thailand | 150 | 0 (0.0) | 1 (0.6) | 149 (99.3) |
| Espino, 2016** [29] | Capillary | Philippines | 302 | 1 (0.3) | 0 (0.0) | 301 (99.7) |
| Espino, 2016** [29] | Venous | Philippines | 302 | 1 (0.3) | 0 (0.0) | 301 (99.7) |
| Henriques, 2018*** [20] | Capillary | Cambodia | 505 | 0 (0.0) | 7 (1.4) | 498 (98.6) |
| Henriques, 2018*** [20] | Capillary | Laos | 757 | 4 (0.6) | 4 (0.6) | 749 (98.9) |
| Oo, 2016 [30] | Venous | Myanmar | 1,000 | 0 (0.0) | 0 (0.0) | 1,000 (100.0) |
| Roca-Feltrer, 2014 [31] | Capillary | Cambodia | 938 | 5 (0.5) | 0 (0.0) | 933 (99.5) |
| Roh, 2016, Uganda [33] | Capillary | Uganda | 645 | 2 (0.3) | 0 (0.0) | 643 (99.7) |
| Satyagraha, 2016 [32] | Venous | Indonesia | 610 | 1 (0.2) | 0 (0.0) | 609 (99.8) |
| von Fricken, 2014 [34] | Venous | Haiti | 456 | 0 (0.0) | 0 (0.0) | 456 (100.0) |
| **Total** | | | **5,815** | **14 (0.2)** | **24 (0.4)** | **5,777 (99.3)** |

*Paired CSG and spectrophotometry results from venous and capillary blood.

**Paired CSG results from venous and capillary blood, spectrophotometry results from venous blood.

***Same publication but different sites.

**Abbreviations:** CSG, CareStart Screening test for G6PD deficiency; G6PD, glucose-6-phosphate dehydrogenase

**Table 2. Details on studies included.**

| Article | Blood | Country | Study population | n With malaria (%)* | Females included (%) | Males included (%) | Calculated local AMM (100% G6PD activity) in U/gHb; G6PD activity at 30%** | n of Study population included with <30% G6PD activity based on local AMM (%) | n of Study population included with <30% G6PD activity based on pooled AMM (%) |
|---|---|---|---|---|---|---|---|---|---|
| Bancone, 2015 [21] | Capillary | Thailand | Healthy volunteers | 0 (0.0) | 95 (68.8) | 43 (31.2) | 6.6; 2.0 | 41 (29.7) | 44 (31.9) |
| Bancone, 2015 [21] | Venous | Thailand | Healthy volunteers | 0 (0.0) | 99 (66.4) | 50 (33.6) | 6.6; 2.0 | 45 (30.2) | 51 (34.2) |
| Espino, 2016 [29] | Capillary | Philippines | High school students from cross-sectional survey | Not provided | 197 (65.5) | 104 (34.6) | 11.1; 3.3 | 17 (5.7) | 16 (5.3) |
| Espino, 2016 [29] | Venous | Philippines | High school students from cross-sectional survey | Not provided | 197 (65.5) | 104 (34.6) | 11.1; 3.3 | 17 (5.7) | 16 (5.3) |
| Henriques, 2018 [20] | Capillary | Cambodia | Participants of cross-sectional survey | Not provided | 248 (49.8) | 250 (50.2) | 7.6; 2.3 | 117 (23.5) | 124 (24.9) |
| Henriques, 2018 [20] | Capillary | Laos | Purposively selected community members | Not provided | 366 (48.9) | 383 (51.1) | 11.5; 3.5 | 39 (5.2) | 38 (5.07) |
| Oo, 2016 [30] | Venous | Myanmar | Healthy volunteers | 0 (0.0) | 476 (47.6) | 524 (52.4) | 8.3; 2.5 | 68 (6.8) | 68 (6.8) |
| Roca-Feltrer, 2014 [31] | Capillary | Cambodia | Adults >18 years, nonpregnant from cross-sectional survey | 0 (0.0) | 484 (51.9) | 449 (48.1) | 12.0; 3.6 | 74 (7.9) | 70 (7.5) |
| Roh, 2016, Uganda [33] | Capillary | Uganda | Children 6–59 months from cross-sectional survey (3.5% with microscopic malaria) | 22 (3.4) | 317 (49.3) | 326 (50.7) | 6.4; 1.9 | 10 (1.6) | 24 (3.73) |
| Satyagraha, 2016 [32] | Venous | Indonesia | All ages from cross-sectional survey (2.5% with malaria) | 15 (2.5) | 349 (57.3) | 260 (42.7) | 9.3; 2.8 | 30 (4.9) | 30 (4.9) |
| von Fricken, 2014 [34] | Venous | Haiti | Primary school children from cross-sectional survey | Not provided | 267 (58.6) | 189 (41.5) | 9.1 | 46 (10.1) | 46 (10.1) |

*Based on publication.

**Calculated cut-offs and cut-offs published in source article do not necessarily match due to different definitions.

**Abbreviations:** AMM, adjusted male median; G6PD, glucose-6-phosphate dehydrogenase

participants were excluded because the spectrophotometry result was missing or had an extreme value (>25 U/gHb), and results from 24 (0.4%) participants were excluded due to an invalid CSG result. A total of 5,777 (99.3%) results were included in the analysis (Table 1), of which 3,095 (53.6%) were from females. The majority of samples were collected from healthy volunteers (Table 2).

## Definition of 100% G6PD activity

In the study with paired spectrophotometry measures of patients with both capillary and venous sampling, there was no significant difference in G6PD activity ($p$ = 0.292) [21]. Results for capillary and venous spectrophotometry were therefore pooled. The site-specific AMM ranged from 6.6 U/gHb to 12.3 U/gHb. When results were pooled across all studies, the overall AMM was 9.2 U/gHb (interquartile range [IQR] 7.2–11.5) (Table 2).

## Pooled performance

The pooled sensitivity was 0.96 (95% CI 0.90–0.99) (Fig 2), and the specificity was 0.95 (95% CI 0.92–0.96) (Fig 3). The number of invalid results was significantly higher for capillary samples (12/3,274) compared to venous samples (2/2,517, $p = 0.022$) (Table 1); the AUC of the SROC was 0.98 (95% CI 0.97–0.99) (S1 Fig).

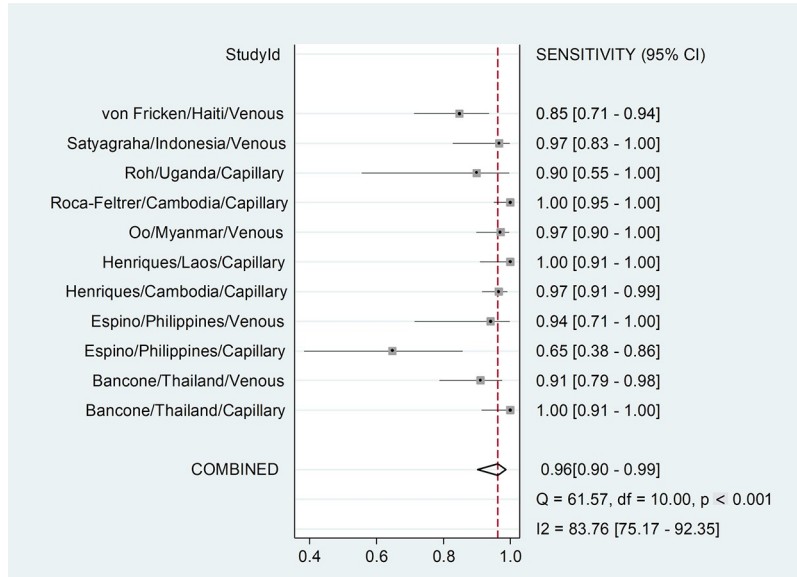

**Fig 2. Forest plot: Sensitivity.** Threshold for G6PDd is calculated based on the site-specific AMM. Study ID is identified by first author, country of sample collection, and type of blood used. AMM, adjusted male median; G6PDd, glucose-6-phosphate dehydrogenase deficiency.

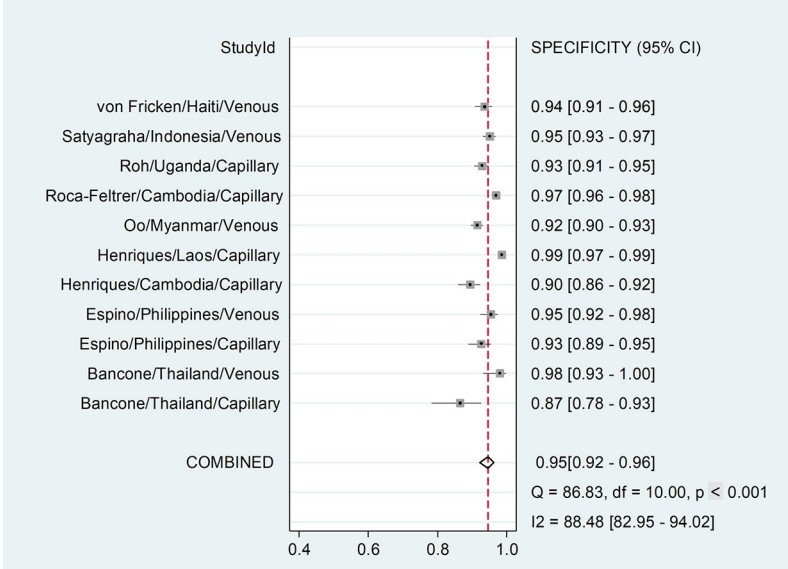

**Fig 3. Forest plot: Specificity.** Threshold for G6PDd is calculated based on the site-specific AMM. Study ID is identified by first author, country of sample collection, and type of blood used. AMM, adjusted male median; G6PDd, glucose-6-phosphate dehydrogenase deficiency.

## Utility of the CSG

When the prevalence of G6PDd was varied from 5% to 30%, the unconditional negative predictive value (NPV) was 0.97 (95% CI:0.94–1.00), and the positive predictive value (PPV) was 0.76 (95% CI 0.72–0.81). The LR+ and LR− were 18.2 (95% CI 13.0–25.5) and 0.05 (95% CI 0.02–0.12), respectively (S2 Fig).

## Publication bias

Three of the 11 eligible studies (enrolling 1,280 participants from Brazil, Yemen, and Ghana) were not included because the corresponding authors did not reply [35–37]. These studies had a higher proportion of malaria patients. The characteristics of studies included and excluded in the individual data analysis are presented in S2 Table. No significant publication bias was detected among included studies (*p* = 0.41); 3 studies were identified as yielding a high risk of bias, 2 due to purposive selection of participants [20,21] and 1 due to lack of temperature-controlled spectrophotometry [34] (Fig 4, S3 Fig).

|  | Risk of bias | | | | Applicability | | |
|---|---|---|---|---|---|---|---|
|  | Patient Selection | Index Test | Reference Standard | Flow and Timing | Patient Selection | Index Test | Reference Standard |
| Roca-Feltrer, 2014 | low | low | low | low | low | low | low |
| Satyagraha, 2014 | low | unclear | unclear | low | low | low | low |
| Bancone, 2015 | high | unclear | unclear | low | low | low | low |
| Espino, 2016 | low | low | low | low | low | low | low |
| Henriques, 2018, Laos | low | low | low | low | low | low | low |
| Henriques, 2018, Cambodia | high | low | low | low | low | low | low |
| Oo, 2016 | low | unclear | unclear | low | low | low | low |
| Roh, 2016 | low | low | low | low | low | low | low |
| von Fricken, 2014 | unclear | low | high | low | low | low | low |

**Fig 4. Qualitative assessment of included studies (QUADAS2).** QUADAS2, Quality Assessment of Diagnostic Accuracy Studies.

## Sensitivity analyses

In the a priori sensitivity analyses, the pooled performance did not vary significantly irrespective of whether capillary or venous blood was collected (*p* = 0.547). For capillary samples, the sensitivity was 0.99 (95% CI 0.80–1.00), and specificity was 0.94 (95% CI 0.90–0.97) compared to 0.93 (95% CI 0.87–0.96) and 0.94 (95% CI 0.92–0.96), respectively, for venous samples. However, performance differed significantly between males and females (*p* = 0.027). In males, the sensitivity was 0.97 (95% CI 0.92–0.99), and specificity was 0.98 (95% CI 0.96–0.99), significantly higher than in females, who had a sensitivity of 0.92 (95% CI 0.80–0.97) and a specificity of 0.93 (95% CI 0.89–0.96) (Table 3).

When 2 studies enrolling purposively selected participants were excluded, the pooled performance was slightly lower (sensitivity 0.95, 95% CI 0.86–0.99; specificity 0.94, 95% CI 0.93–0.96) [20,21]. When one study using venous samples in which spectrophotometry was not temperature controlled was excluded, the overall pooled performance was unchanged (sensitivity 0.96, 95% CI 0.89–0.98; specificity 0.95, 95% CI 0.93–0.96), although the performance for venous samples was slightly lower (sensitivity 0.95, 95% CI 0.89–0.98; specificity 0.95, 95% CI 0.92–0.97).

When the analysis was repeated using an AMM derived from pooled spectrophotometry data rather than the site-specific AMM, the pooled performance did not differ (sensitivity 0.96, 95% CI 0.89–0.98; specificity 0.95, 95% CI 0.93–0.96) (Table 3 and S4 Fig). When aggregated data were included from the 3 studies that fulfilled the inclusion criteria, but for which no IPD were available, the performance did not change (Table 3, S5 Fig).

**Table 3. Results of pooled and sensitivity analysis.**

| Analysis | Sensitivity (95% CI) | Specificity (95% CI) | Sample size |
|---|---|---|---|
| Primary analysis | 0.96 (0.90–0.99) | 0.95 (0.92–0.96) | 5,777 |
| Capillary only | 0.99 (0.80–1.00) | 0.94 (0.90–0.97) | 3,262 |
| Venous only | 0.93 (0.87–0.96) | 0.94 (0.90–0.97) | 2,515 |
| Males only | 0.97 (0.92–0.99) | 0.98 (0.96–0.99) | 2,682 |
| Females only | 0.92 (0.80–0.97) | 0.93 (0.89–0.96) | 3,095 |
| Excluding studies with purposively selected participants | 0.95 (0.86–0.99) | 0.94 (0.93–0.96) | 4,243 |
| Excluding studies without temperature-controlled spectrophotometry | 0.96 (0.89–0.98) | 0.95 (0.93–0.96) | 5,321 |
| Applying pooled AMM | 0.96 (0.89–0.98) | 0.95 (0.93–0.96) | 5,777 |
| Considering aggregate data from all eligible studies | 0.96 (0.90–0.99) | 0.95 (0.92–0.96) | 7,057 |

**Abbreviation:** AMM, adjusted male median

## Discussion

In this meta-analysis, we observed an overall sensitivity and specificity of the CSG of more than 95%; the NPV was almost 100% across a wide range of G6PDd prevalences. The high LR+ and low LR− suggest that the CSG is suitable for confirmation as well as exclusion of G6PDd at a 30% threshold level.

The CSG performed significantly better in males compared to females. The CSG performs best at an approximate 30% cut-off activity [20]; however, the absolute cut-off of the CSG and the absolute cut-off calculated from spectrophotometry do not necessarily match. Since males are either hemizygous normal or deficient, their enzyme activity will be either below or well above the 30% cut-off, and small discrepancies between the 2 thresholds will not affect the calculated performance. However, females can be either homozygous or heterozygous for the G6PD gene, the latter manifesting phenotypically with enzyme activities ranging from almost normal to G6PD deficient [12–14]. Therefore, in heterozygous females, small differences between the inherent test and the calculated cut-off activity will affect the test's performance adversely.

The performance of the CSG was slightly better in samples collected from capillary compared to venous blood, although this did not reach statistical significance. However, the overall performance was more reliable in studies using venous blood, which had a lower number of invalid results. Bancone and colleagues previously compared CSG results from paired venous and capillary samples, with 11% discrepancy between samples, with a sensitivity at the 30% threshold of 100% in capillary samples compared to 89% in venous samples [21]. In the same study, the authors also correlated their findings with haematological parameters and found that RBC concentration, haemoglobin, haematocrit, mean corpuscular volume, and platelet count varied slightly between venous and capillary samples; however, they concluded that these differences were unlikely to have a major effect on the performance of the CSG [21]. In contrast, a study conducted by Espino and colleagues reported lower sensitivities for diagnosing deficiency at the 30% threshold among capillary samples (69% sensitivity) compared to their paired venous counterparts (94% sensitivity) [29].

Despite the significantly higher number of invalid results, the CSG is more likely to be performed on capillary blood from a finger prick, following the same procedures as for malaria rapid diagnostic tests. The observed good performance of the CSG on capillary blood is therefore reassuring; the pooled sensitivity is similar to the widely used FST [38]. While the CSG and the FST can be applied to screen patients for G6PDd prior to administering PQ, the recommended criteria for the recently licensed 8-aminoquinoline drug tafenoquine (TQ) are more stringent and require diagnosis of G6PDd at a 70% threshold, which requires a quantitative assay [39,40].

In reality, G6DP testing is rarely available in malaria-endemic communities, and therefore PQ is often not prescribed due to fear of inducing haemolysis in vulnerable patients [2]. The availability of a robust point-of-care G6PD test to screen patients prior to treatment provides a significant advance that will enhance the uptake of radical cure into routine practice. Unfortunately, the CSG does not have a control line, and this has implications for implementation into routine practice. Previous studies have shown that, at a cost of US$1.75, the use of the CSG is a cost-effective strategy at enhancing safe and effective radical cure with PQ [41].

### Limitations

Our study has a number of limitations. The geographical spread of results included was limited, with most studies being conducted in Southeast Asia. It is likely that the performance, including PPV and NPV of the tests, will vary with the local context, including the prevalence and variants of G6PDd and the training and education of the clinic staff.

Only a few data variables were collated from all studies, and therefore our covariate analysis was limited to the haemoglobin concentration, the sex of the participant, and the country of sample collection. Other factors that may also have influenced the test results could have included batch to batch variability in test kits, the temperature at which the tests were performed, and training and ability of individuals undertaking the tests.

Spectrophotometry remains the gold standard for the diagnosis of G6PDd and was used as the reference for the current analysis [16]. Alternative approaches, such as molecular analysis for G6PD variants correlate poorly with G6PD phenotype, precluding use of this approach as reliable reference [33,42–44]. In a comparison between Trinity spectrophotometry kits, considered for this analysis, and another spectrophotometry kit (Pointe Scientific, Canton, MI), both assays showed a very good correlation (r = 0.9799, $p < 0.001$) [45].

The AMM was calculated for each site specifically; consequently, the absolute cut-off activity in U/gHb of the reference method spectrophotometry varied across sites. To assess whether this had an impact on the pooled performance, the analysis was repeated calculating a universal AMM across all sites; reassuringly, the results of the pooled performance did not differ.

IPD from 3 eligible studies, enrolling 1,280 participants, were not available [35–37]. In contrast to the included studies, the proportion of malaria patients among the excluded studies was higher. It is possible that malaria influences G6PD activity, although it is unlikely that this would have impacted the observed performance because CSG and spectrophotometry testing were done on the same sample.

Reassuringly, when the analysis was repeated including aggregated data, the test performance did not change. Finally, all studies included were performed under research conditions and by well-trained study staff; in real-life settings, the performance of the CSG could be lower.

### Conclusion

The results from this pooled analysis suggest that the CSG provides a reliable method to identify individuals with less than 30% G6PD enzyme activities; based on these findings, the test is suitable for introduction into routine treatment prior to PQ but not TQ treatment. Further operational research is required to assess how the test performs under real-life conditions.

### Supporting information

**S1 PRISMA Checklist. PRISMA IPD checklist.** PRISMA, Preferred Reporting Items for Systematic reviews and Meta-Analyses.
(DOCX)

**S1 Table. Test methods applied.**
(DOCX)

**S2 Table. Details on studies not included.**
(DOCX)

**S3 Table. Contact details from which data were obtained.**
(DOCX)

**S1 Fig. Summary ROC.** ROC, Receiver Operating Characteristics curve.
(TIF)

**S2 Fig. Likelihood ratio scatter diagram.**
(TIF)

**S3 Fig. Deeks' funnel plot asymmetry test.**
(TIF)

**S4 Fig. Forest plot: Sensitivity and specificity, pooled AMM.** Threshold for G6PDd is calculated based on the pooled AMM.
(TIF)

**S5 Fig. Forest plot: Sensitivity and specificity, all studies fitting the inclusion criteria.** Threshold for G6PDd is based on definitions provided from aggregate data (first 3 studies from top) and calculated based on the site-specific AMM for IPD (all other studies). Study ID is identified by first author, country of sample collection, and type of blood used.
(TIF)

## Acknowledgments

We would like to thank all participants of the studies included, as well as all staff that contributed to the primary articles included. We would also like to thank Mr. Sharif Hossain, who provided statistical advice.

## Author Contributions

**Conceptualization:** Benedikt Ley, Ari Winasti Satyagraha.

**Data curation:** Benedikt Ley, Ari Winasti Satyagraha, Hisni Rahmat.

**Formal analysis:** Benedikt Ley, Amalia Karahalios.

**Investigation:** Benedikt Ley.

**Methodology:** Benedikt Ley.

**Project administration:** Benedikt Ley.

**Supervision:** Benedikt Ley.

**Validation:** Benedikt Ley, Ari Winasti Satyagraha, Hisni Rahmat, Michael E. von Fricken, Nicholas M. Douglas, Daniel A. Pfeffer, Fe Espino, Lorenz von Seidlein, Gisela Henriques, Nwe Nwe Oo, Didier Menard, Sunil Parikh, Germana Bancone, Amalia Karahalios, Ric N. Price.

**Visualization:** Benedikt Ley.

**Writing – original draft:** Benedikt Ley.

**Writing – review & editing:** Benedikt Ley, Ari Winasti Satyagraha, Hisni Rahmat, Michael E. von Fricken, Nicholas M. Douglas, Daniel A. Pfeffer, Fe Espino, Lorenz von Seidlein, Gisela Henriques, Nwe Nwe Oo, Didier Menard, Sunil Parikh, Germana Bancone, Amalia Kara-halios, Ric N. Price.

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
