## [Decision Letter · Decision Letter 0]

26 Sep 2019

Dear Dr. Ley,

Thank you very much for submitting your manuscript "A meta-analysis of the performance of the Carestart™ rapid diagnostic test for the detection of glucose 6 phosphate dehydrogenase deficiency" (PMEDICINE-D-19-02311) for consideration at PLOS Medicine. 

Your paper was evaluated by a senior editor and discussed among all the editors here. It was also discussed with an academic editor with relevant expertise, and sent to three independent reviewers, including a statistical reviewer. The reviews are appended at the bottom of this email and any accompanying reviewer attachments can be seen via the link below:

[LINK]

In light of these reviews, I am afraid that we will not be able to accept the manuscript for publication in the journal in its current form, but we would like to consider a revised version that addresses the reviewers' and editors' comments. Obviously we cannot make any decision about publication until we have seen the revised manuscript and your response, and we plan to seek re-review by one or more of the reviewers. 

We expect to receive your revised manuscript by Oct 17 2019 11:59PM. Please email us (plosmedicine@plos.org) if you have any questions or concerns.

We look forward to receiving your revised manuscript. 

Sincerely,

Thomas McBride, PhD

Senior Editor 

PLOS Medicine

plosmedicine.org

1- In addition to the data statement, it would be helpful to provide a list of the data contacts for the included studies in a supplemental file.

2- Please revise your Title to PLOS Medicine style, placing the study design at the end, after a colon “Performance of the Carestart rapid diagnostic test for the detection of glucose 6 phosphate dehydrogenase deficiency: a Systematic review and meta-analysis”

3- PLOS style does not permit trademarks, please remove the (TM) in the title and throughout the manuscript.

4- Please combine the Abstract “Methods” and “Results” sections into one section, titled “Methods and Results”. 

5- Please include the full dates (day, month, year) of the search in the Abstract, as well as the data sources, eligibility criteria, and synthesis/appraisal methods

6- Alongside the 95% CIs, please include p-values.

7- In the last sentence of the Abstract Methods and Findings section, please describe the main limitation(s) of the study's methodology.

8- At this stage, we ask that you include a short, non-technical Author Summary of your research to make findings accessible to a wide audience that includes both scientists and non-scientists. The Author Summary should immediately follow the Abstract in your revised manuscript. This text is subject to editorial change and should be distinct from the scientific abstract. Please see our author guidelines for more information: https://journals.plos.org/plosmedicine/s/revising-your-manuscript#loc-author-summary

9- Please update your search to the present time.

10- Please evaluate study quality and risk of bias.

11- Line 97, I believe you mean “sex” rather than “gender”. Please make sure the usage is accurate.

12- In the first paragraph of the Discussion and in the Discussion conclusions, please address the study implications without overreaching what can be concluded from the data; the phrase "In this study, we observed ..." may be useful.

Comments from the reviewers:

Reviewer #1: Thank you for the opportunity to review this interesting paper. This study presented an IPD meta-analysis of the diagnostic accuracy of the Carestart test for the detection of glucose 6 phosphate dehydrogenase deficiency (GPDDd) which showed good accuracy (according to diagnostic test metrics). The clinical message is simple but rather important - this is a rapid diagnostic test which can be deployed cheaply in the field to identify to GPDDd in individuals with malaria prior to being treated with primaquine to prevent primaquin-inducted can cause haemolysis. The methods are appropriate and I commend the authors for using an IPD approach for a diagnostic test accuracy study which can be challenging to synthesize. The methods are appropriate. The authors have utilised the appropriate approach of a two level model with a bivariate normal model with a logit transformation of sensitivity and specificity between studies. See https://www.ncbi.nlm.nih.gov/pubmed/18816508 - this may be a useful reference for the authors. The results are presented clearly and succinctly, and the conclusions support the results. Thus, I have only a few minor suggestions:

- Lines 133-136: the explanation of likelihood ratios is not very clear and they are described as probabilities in the text. LR+ and LR- are not probabilities - they are ratios, correct interpretation: LR+: quantifies how much more likely the positive test of G6PDd is to occur in subjects with the condition compare those without; LR-: quantifies how much less likely the negative test of G6PDd will occur in subjects without disease 

- Lines 136-137: this statement needs be benchmarked to accepted criteria. see - https://www.ncbi.nlm.nih.gov/pmc/articles/PMC4975285/. Generally rule is that good test have LR+ > 10 and LR- < 0.1. The results show that the Carestart test satisfies this benchmark. 

- Lines 194-196: the authors excluded three studies from the IPD due to not providing study data. Given that three studies did not reply but did meet eligibility criteria, it would be helpful to consider pooling the aggregate study reported outcomes together for 11 studies. This will allow for a comparison with the aggregate samples, given that the three eligible studies contribute about 1280 participants with a higher proportion of malaria patients. That's quite a large number of patients to not include in any of the analyses. 

- Discussion section: It would be helpful if the authors could benchmark the Carestart diagnostic accuracy results from their IPD against the standard approach of mass spec from the literature. It would also be helpful for the authors to state if there are any of comparable reviews in this area and if so, how do their results compare. If not, they should also state the novelty of their current study. 

- Limitations section: understand there was a limited data request from the study authors to data providers and hence there were a limited number of covariates. The authors should address this issue as a limitation, and speak to if any potential covariates which have not been captured could influence the results. 

Reviewer #2: A timely and useful analysis. A few minor remarks for the authors to consider. 

1. Lines 63, 64. This seems unhelpfully vague. What are the possible clinical consequences? Also, "can cause"? This seems to infer inconsistency of the effect, i.e., sometimes it does not. If that's true, say so and cite the evidence. Otherwise, the evidence available would lead this reviewer to believe "invariably causes" to be the more appropriate terminology.

2. Line 65. "More than 215"? How many more? This seems a peculiar expression given the precision of the number. 

3. Line 72. They are certainly at significant risk simply by the genetics. The authors perhaps mean "at risk of significant harm". 

4. Line 77. The expressed affordability of the FST may be challenged. It is about $5 a test just for the reagents and requires a cold-chain and specialised equipment. It is certainly not sufficiently affordable to even referral hospitals in endemic zones, where it is almost never found (because it costs too much). 

5. Line 226-227. Doubtful that the CSG was engineered to perform poorly above 30% of normal, i.e., "designed". It is likely to be an inherent limitation of the technology. 

Reviewer #3: This is a metadata analysis on field evaluations of a G6PD deficiency (G6PDd) point-of-care (POC) RDT CareStart produced by AccessBio. This analysis included 8 publications, which had both RDT and quantification of G6PD activity. Given that the use of primaquine and tafenoquine for radical cure of vivax malaria requires the screening of G6PDd, the need for a reliable POC device to detect G6PDd is important. Overall, the analysis is well done. Since one aim of the journal is to have an impact on medical practice, it would be important not to include a lot of statistical jargons without explanation in the abstract.

1. The authors used unconditional negative predictive value (NPV), positive (LR+) and negative (LR-) likelihood ratios in the abstract. However, for the general audience, they are not immediately understandable (though they are explained in the main text). Suggest to include a short description of what the meanings of the results are.

2. Line 76: the gold standard quantitative method is expensive - it should be clarified that there are many G6PD quantitative kits available, but the one from Trinity is much more expensive.

3. Some studies compared the assay results from both venous and capillary blood. I wonder whether there is any basis suggesting that the two sources would be different in G6PD activity? 

4. In sensitivity analysis, the sensitivity for males was significantly higher than for females. Is there a potential explanation for this?

Some editorial suggestions:

1. For consistency with published literature, please use PQ as abbreviation for primaquine and use it throughout the text (see lines 74, 254).

2. I suggest to use CareStart in the abstract, since it is difficult to see why Carestart is abbreviated into CSG.

3. In the abstract Introduction section, "a qualitative lateral flow assay from Carestart, USA" should be "CarestartTM G6PD RDT, a qualitative lateral flow assay from AccessBio, USA".

4. Line 222 , use abbreviation NPV or spell out it in the entire text.

[LINK]

---

## [Editor Report · Decision Letter 1]

25 Oct 2019

Dear Dr. Ley,

Thank you very much for re-submitting your manuscript "Performance of the Accessbio/Caretstart rapid diagnostic test for the detection of glucose-6-phosphate dehydrogenase deficiency: a systematic review and meta-analysis" (PMEDICINE-D-19-02311R1) for review by PLOS Medicine.

I have discussed the paper with my colleagues and the academic editor. I am pleased to say that provided the remaining editorial and production issues are dealt with we are planning to accept the paper for publication in the journal.

[LINK]

Please also check the guidelines for revised papers at http://journals.plos.org/plosmedicine/s/revising-your-manuscript for any that apply to your paper. 

We look forward to receiving the revised manuscript by Nov 01 2019 11:59PM. 

Sincerely,

Thomas McBride, PhD

Senior Editor 

PLOS Medicine

plosmedicine.org

Requests from Editors:

1- Please add this statement to the manuscript's Competing Interests: "LVS receives a stipend as a Specialty Consulting Editor for PLOS Medicine and serves on the journal's editorial board."

2- Thank you for providing the contact information for accessing data from the primary studies in table S3. However, PLOS does not allow authors of the current manuscript to be the primary contact for data access. Please make these data available (either in the supplemental files or in a data repository) or provide an alternate contact for data access.

3- Apologies for my mistake, please rename the Abstract “Methods and results” section “Methods and findings”. Please also rename the Abstract “Discussion and conclusion” section “Conclusions”.

4- In the Abstract, please briefly note the regions giving rise to the studies and at least broad details of the participants (adults vs children).

5- Was the start date for the pubmed search Jan 1, 2011? Please specify the month and date in the Abstract and the Methods.

6- Thank you for including an author summary. It may be useful to readers to explain the significance of the 30% cutoff for G6PD activity. 

7- In your response to Reviewer 1, point 1, it may be more clear to state: “. In the case of the CSG the positive likelihood ratio (LR+) describes how many times more likely a G6PDd test result is to occur in a G6PDd individual compared to in a G6PD normal individual. The negative likelihood ratio (LR-) is defined as the inverse of this, or how much less likely the negative test of G6PDd is to occur in a G6PD normal person compared to a G6PDd individual (26).”

8- Please preface the description of the analysis that includes aggregate data from eligible articles without IPD with “In response to a reviewer request…” or similar.

9- The limited geographical span of the studies could be listed as a limitation, both in the Abstract and the Discussion.

10- There is a p= 0.00 in the forest plot. Please report as p < 0.001.

11- Line 336 (of the marked up manuscript): “*In a comparison between* Trinity spectrophotometry kits, considered for this analysis, *and* another spectrophotometry kit…”. To make clear that the comparison was not part of the current study.

Comments from the Academic Editor:

Line 53: should be "95%CI" not "955CI"

Line 97: Use "haemolysis" to be consistent with the British spelling (the manuscript mostly used British spelling)

Line 86: spell out Plasmodium for both species

Line 147, 291: use "PQ" for primaquine since it is abbreviated earlier (check the entire text)

Line 218-226: all "95%CI" were written as "95CI"

Line 220: use AUC since it was abbreviated earlier

Line 265-267: this sentence does not read well, suggest to revise (remove "and thus"?)

Line 278: use "haem" for all three words in this line

[LINK]

---

## [Editor Report · Decision Letter 2]

8 Nov 2019

Dear Dr Ley, 

On behalf of my colleagues and the academic editor, Dr. Liwang Cui, I am delighted to inform you that your manuscript entitled "Performance of the Accessbio/Caretstart rapid diagnostic test for the detection of glucose-6-phosphate dehydrogenase deficiency: a systematic review and meta-analysis" (PMEDICINE-D-19-02311R2) has been accepted for publication in PLOS Medicine. 

PRODUCTION PROCESS

PRESS

PROFILE INFORMATION

Thank you again for submitting the manuscript to PLOS Medicine. We look forward to publishing it. 

Best wishes, 

Thomas McBride, PhD

Senior Editor 

PLOS Medicine

plosmedicine.org